# Processing of D1 Protein: A Mysterious Process Carried Out in Thylakoid Lumen

**DOI:** 10.3390/ijms23052520

**Published:** 2022-02-25

**Authors:** Noritoshi Inagaki

**Affiliations:** Research Center for Advanced Analysis, National Agriculture and Food Research Organization (NARO), Tsukuba 305-8518, Japan; ninagaki@affrc.go.jp

**Keywords:** assembly, carboxyl-terminal extension, carboxyl-terminal processing protease (CtpA), D1 protein (*psbA*), Mn_4_CaO_5_ cluster (Mn-cluster), oxygen evolution, photosystem II (PSII), serine-lysine dyad, thylakoid lumen, water oxidation

## Abstract

In oxygenic photosynthetic organisms, D1 protein, a core subunit of photosystem II (PSII), displays a rapid turnover in the light, in which D1 proteins are distinctively damaged and immediately removed from the PSII. In parallel, as a repair process, D1 proteins are synthesized and simultaneously assembled into the PSII. On this flow, the D1 protein is synthesized as a precursor with a carboxyl-terminal extension, and the D1 processing is defined as a step for proteolytic removal of the extension by a specific protease, CtpA. The D1 processing plays a crucial role in appearance of water-oxidizing capacity of PSII, because the main chain carboxyl group at carboxyl-terminus of the D1 protein, exposed by the D1 processing, ligates a manganese and a calcium atom in the Mn_4_CaO_5_-cluster, a special equipment for water-oxidizing chemistry of PSII. This review focuses on the D1 processing and discusses it from four angles: (i) Discovery of the D1 processing and recognition of its importance: (ii) Enzyme involved in the D1 processing: (iii) Efforts for understanding significance of the D1 processing: (iv) Remaining mysteries in the D1 processing. Through the review, I summarize the current status of our knowledge on and around the D1 processing.

## 1. Introduction

Oxygenic photosynthesis is the most crucial energy source available for organisms on planet Earth. Considering not only fossil fuels but also biofuels, oxygenic photosynthesis is also an indispensable bioprocess for the emergence, prosperity and sustainability of modern civilization [1,2]. Primary light reactions in the oxygenic photosynthesis are carried out on two photosystems, photosystem I and II (hereafter PSI and PSII, respectively) which are large pigment-protein complexes embedded in thylakoid membranes [1,3]. The PSII is distinctive that it has capacity to produce strong oxidant at its reaction center chlorophyll pair (P680). The P680 on excitation state undergoes charge separation and redox potential of resultant P680^+^ is estimated at approximately 1.2 V sufficient to oxidize water molecules [3,4,5]. To link one-electron charge separation to four-electron chemistry of water oxidation, a special equipment, the water-oxidizing complex (WOC) is formed on lumen side of PSII, which is composed of a distorted chair form consisting of four atoms of manganese, one atom of calcium and five atoms of oxygen (Mn_4_CaO_5_ cluster) [5,6]. Through the water oxidation on PSII, molecular oxygen is evolved as a by-product, which completely changed the global environment oxidatively. This drastic change termed Great Oxidation Event has encouraged appearance and prosperity of aerobic organisms, including us [7].

The PSII core complex consists of at least 20 protein subunits, photosynthetic pigments, quinones and metal atoms. Among them, D1 protein (*psbA* gene product) and its homologous partner, D2 protein (*psbD* gene product) form a central pillar of the PSII complex [6,8]. Despite the functional importance of the D1 protein, this subunit is known to be unstable in the light [9,10]. Reactive oxygen species (ROS) inevitably generated by normal function of PSII [11], might specifically damage the D1 protein. The decrease of PSII activity caused by the light-induced damage of D1 protein has been studied as photoinhibition for long time [12,13], however there is no consensus on the molecular mechanism to damage the D1 protein exclusively. On the other hand, the repair process from the damaged PSII is clearer. It consists of the following two stages: the damaged D1 proteins are removed from the PSII and newly synthesized D1 proteins are assembled with the rest of PSII subunits on thylakoid membranes [14,15,16] (Nota bene 1: Although it is necessary to consider de novo biosynthesis and repair process separately, this review focuses on the repair process of PSII that is more frequently performed on thylakoid membranes in the light).

Proteases mediate a wide range of regulational mechanisms in cells, not only cleaning up injured proteins for keeping homeostasis. Especially, processing protease/peptidase is involved in proteolytic maturation of the target proteins to express each function. Several processing proteases are known to act in chloroplasts for construction, regulation or maintenance of photosynthetic equipment including PSII [17,18,19].

The D1 protein is synthesized as a precursor form (hereafter pD1) which has a small extension at its carboxyl-terminus (C-terminus). Maturation of D1 protein is achieved with a proteolytic removing of the extension by a specific processing protease, CtpA [20]. Despite a seemingly simple step, the D1 processing has profound significance in construction of the WOC. Moreover, several mysterious issues are remaining in and around the D1 processing. In this review, I focus on the D1 processing and would like to trace the history of its discovery. I also introduce progress of the research and raise the unsolved issues in this research area.

## 2. Discovery of D1 Processing and Recognition of Its Importance

Due to the influence of the complicated research history, D1 protein had some more names. To draw information of the D1 protein from historical literatures, the readers need to understand these names. At least, D1 protein was also called atrazine-binding protein, herbicide-binding protein, Q_B_ protein, peak D, photogene 32 product or 32 kDa (or 32,000- dalton) protein, which names were derived from each focus of the research.

### 2.1. Discovery of the D1 Processing

Prior to the recognition that the D1 protein is a core subunit of PSII [8], the massive biosynthesis of this protein on thylakoids in the light was an eye-catching phenomenon and well analyzed [21,22]. In late 1970s, several researchers noticed in their pulse-chase experiments that newly synthesized D1 protein (a 32,000 Dalton protein in these references) is slightly larger than that enough time has passed [23,24]. As expected, the larger molecular species is pD1 [25], and subsequent precise peptide-mapping analyses revealed that the pD1 has a C-terminal extension [26]. C-terminus of mature D1 protein of spinach is Ala344 determined by direct sequencing [27,28], which revealed that C-terminal extension of spinach consists of 9 residues, based on the sequence deduced from the *psbA* gene [29]. Soon after, the C-terminus of D1 protein of *Synechocystis* sp. PCC6803 (*Synechocystis* 6803) was determined to be Ala344. However, the C-terminal extension of this organism consists of 16 residues long [30]. Accumulated sequence data by comprehensive genome analyses of oxygenic photosynthetic organisms have revealed that lengths of the C-terminal extensions have some variations roughly classified into three groups [20], major members of the group I are chlorophyll *b* containing organisms (C-terminal extensions of this group consist of 8-9 residues), major members of the group II are most cyanobacteria and red lineage photosynthetic eukaryotes holding primary plastids [31,32,33,34] (C-terminal extensions of this group consist of 15-16 residues) and major members of the group III are some photosynthetic eukaryotes holding secondary or tertiary plastids [31,32,33,34] (Organisms in this group do not have C-terminal extension). Evolution of C-terminal extensions in the group I and II will be considered in Section 5.1, and extension-less organisms in the group III will be discussed in Section 4.1.

### 2.2. Contribution of LF-1 Mutant to Recognize Importance of the D1 Processing

The properties of chlorophyll fluorescence provide valuable information for probing the state of photosynthetic equipment. Taking advantage of its non-destructive feature, it has been frequently used for the isolation of various photosynthetic mutants [35]. LF-1 mutant of a green alga *Scenedesmus obliquus*, which was isolated as a low fluorescent yield mutant [36,37], has expanded our knowledge about physiological importance of the D1 processing. When an external electron donor, diphenylcarbazide, is provided, the LF-1 mutant emits normal chlorophyll fluorescence and indicates normal PSII activity (as measured by dichlorophenolindophenol photoreduction) [36]. These results indicate that the mutant is restrictedly deficient in water-oxidizing capacity of PSII. This interpretation is consistent with a fact that content of Mn, an integral metal atom for water oxidation, is less than half in the LF-1 thylakoids compared with that in the wild-type thylakoids [36]. In addition, redox active Mn could not be detected by EPR (electron paramagnetic resonance) analysis of PSII enriched membrane of the LF-1 mutant [38].

Metz et al. [36,37,39,40] found that the D1 protein in this mutant displays a slightly higher molecular mass and they noticed that the observation is closely related with the reports in which the newly synthesized D1 protein (32,000 Dalton protein) is slightly larger in the pulse chase experiments [23,24]. As they expected, this larger species detected in the LF-1 mutant was confirmed to be pD1 [41,42]. PSII enriched membranes from the LF-1 mutant were treated with Triton X-100 extracts from wild-type thylakoids, which resulted in a specific reduction in molecular mass on the LF-1 D1 protein, like the D1 processing. In addition, oxygen-evolving (water-oxidizing) capacity of LF-1 membranes was photoactivated after the treatment [42]. These early works established that the LF-1 mutant lacks the processing protease for D1 processing and the pD1 in the LF-1 membranes directly prevents construction of the WOC of PSII (Nota bene 2: Currently, reinstatement of the scientific name “*Tetradesmus*” has been proposed [43], and thus readers should pay attention to some reports that describe “*Scenedesmus*” as “*Tetradesmus*”).

## 3. Enzyme Involved in the D1 Processing

Results of the LF-1 mutant suggest that there are two approaches to identify the enzyme(s) involved in this process. One is genetic identification of gene(s) encoding the enzyme(s) from WOC deficient mutants. The other is the biochemical approach to purify enzyme(s) involved in the process based on the activity from extracts of thylakoids.

### 3.1. Genetic Identification of the D1 Processing Protease

Since the 1980s, molecular biological analyses using *Synechocystis* 6803 have developed extensively [44,45,46]. Under the situation, Shestakov et al. [47] isolated a spontaneous *Synechocystis* 6803 mutant indicating deficiency of photosynthesis attributed to blocking at water-oxidizing side of PSII. The mutant was complemented with a genome fragment of *Synechocystis* 6803 carrying a gene (slr0008) encoding a 46.7 kDa protein, homologous with an *Escherichia coli* protease, Prc/Tsp (processing involving the C-terminal cleavage/tail-specific protease) [48,49]. Inactivation of the gene (slr0008) resulted destruction of water-oxidizing activity and detection of pD1 on the thylakoids [50]. The results from forward and reverse genetics experiments indicate that the isolated gene encodes the processing protease of D1 protein and named *carboxyl-terminal processing protease A* (*ctpA*) [47,50]. Based on sequence similarity, putative orthologous genes of spinach and barley were cloned [51].

### 3.2. Biochemical Identification of the D1 Processing Protease

The most critical point for biochemical purification of the enzyme is establishment of an easy procedure to measure its activity. pD1 on the LF-1 thylakoids can be used as a substrate for the protease. D1 processing level on the LF-1 thylakoids can also be evaluated by photoactivation competence [42]. On the other hand, in vitro translated pD1 from spinach chloroplasts RNA can be used as a substrate for the protease [52]. Similarly, in vitro transcribed RNA from SP6 promoter driven *psbA* gene can be applied for preparation of the substrate [53]. Moreover, synthetic peptides having C-terminal sequence of the pD1 can also be used as a substrate for the protease [54,55].

As a result of purification, the spinach protease involved in the D1 processing was isolated as a 45 kDa monomeric protein [53]. Using the amino acid sequences determined from the purified protease, the gene for the protease was cloned from a spinach cDNA library and sequenced [56]. Soon after, the *Schenedesmus* protease was purified and sequenced [57]. Finally, the CtpA proteases have been independently identified from the three organisms (*Synechocystis* 6803, spinach and *Schenedesmus*), these deduced amino acid sequences are highly homologous each other [57].

### 3.3. Homologous Proteases of the CtpA

Homologous proteases of the CtpA were detected in prokaryotes and plants. In non- photosynthetic bacteria, these proteases mediate various cell regulatory mechanisms, such as DNA damage-induced SOS response in *Staphylococcus* [58], virulence factor production in *Bordetella* [59] and sporulation regulation in *Bacillus* [60,61], moreover functional studies are actively conducted on these proteases.

In the oxygenic photosynthetic prokaryotes, several paralogous genes of the *ctpA* were usually detected on the genome. In *Synechocystis* 6803, in addition to the *ctpA* (slr0008), two other paralogous genes, named *ctpB* (slr0257) and *ctpC* (slr1751) were identified on the genome [62,63]. These two genes are not considered to be involved in the D1 processing, because the D1 processing ability has been completely lost in the *ctpA* deficient mutant [47,50]. Exact function of these paralogs has not been clarified, but the inactivation of the *ctpC* provides lethality, implying that the *ctpC* should act in crucial housekeeping processes directly linking with survival [62]. In the oxygenic photosynthetic eukaryotes, several homologous gene sequences have been deposited on the databases, however, the characterization of these homologs except for orthologs of the CtpA has not been progressed.

### 3.4. Localaization of the CtpA

Before the structural data of PSII was obtained [6], D1 protein was thought to be a thylakoid-embedded protein. Trebst [64] farsightedly proposed a topology model of PSII D1/D2 in thylakoid membrane, based on the analogy with the bacterial L/M reaction center [65]. In this model (this model is still basically reasonable), D1 protein has five transmembrane helices and its C-terminus is exposed to the lumen side of thylakoids (Figure 1). This topology means that the nucleus-encoded eukaryotic CtpAs have to be imported into the correct location through endogenous mechanisms, such as chloroplast import and thylakoid translocation [66,67,68]. In general, proteins destined for thylakoid lumen have a bipartite pre-sequence at their amino-terminus (N-terminus) which consists of a transit peptide for translocation into chloroplasts and a signal peptide for targeting thylakoid lumen [67]. Each part of the bipartite pre-sequences is removed in order, by the stromal processing protease and by the thylakoidal processing protease [18]. As expected, spinach CtpA holds a 150 residues long pre-sequence at its N-terminus [51,56] and *Scenedesmus* CtpA has also 77 residues long N-terminal pre-sequence [57]. Since there is no decisive motif that defines the sequence of the transit peptide for translocation into chloroplasts, we do not have definitive evidence that the first half of the pre-sequence is a chloroplast transit peptide. In contrast, the signal peptide for targeting thylakoid lumen is characterized by the distribution of basic residues and the hydrophobicity of the sequence, and these properties are detected in the last 40 residues of both pre-sequences (Figure 2). My experience that intact thylakoid protected the CtpA from external protease treatment, suggest that the CtpA acts in the thylakoid lumen. Based on the interpretation, these pre-sequences probably mediate translocation of the CtpA into the thylakoid lumen of the chloroplasts. On the other hand, *Synechocystis* 6803 CtpA, a prokaryotic protease, has a 31 residues long N-terminal pre-sequence (Figure 2), which possesses a characteristic for targeting thylakoid lumen only [47,69].

Targeting mechanism of thylakoid lumen consists of at least two pathways, the general secretory (Sec) pathway and the twin-arginine translocation (Tat) pathway [66]. The characteristics of the pre-sequences for the targeting of thylakoid lumen of *Synechocystis* 6803 CtpA are enigmatic, because the sequences possess twin arginine residues at the key position for the Tat characteristic, but the whole sequences are likely to be similar with Sec-consensus (Figure 2). To evaluate which pathway is used for import of the CtpA into thylakoid lumen, Karnauchov et al. [69] designed in organelle experiments using inhibitors and competitor proteins for each pathway. Inhibition/saturation of one pathway could not completely repress the permeation of the CtpA into thylakoid lumen. Further quantitative analysis indicated that the CtpA is transported predominantly by the Sec pathway, but incorporated partly through the Tat pathway, meaning that the *Synechocystis* 6803 CtpA is imported into thylakoids by the two pathways in parallel. However, it should be noted that this result was acquired by a chimeric design experiment consisting of the cyanobacterial signal sequence and spinach endogenous factors for thylakoid import, therefore further studies under the natural combination are required for rigorous confirmation.

Since twin arginine motif could not be detected upstream of the hydrophobic region on the pre-sequences of spinach and *Scenedesmus* CtpAs (Figure 2), targeting of thylakoid lumen of these CtpAs might be performed through the Sec pathway. In any case, it is clear that CtpA acts in the thylakoid lumen after removal of the pre-sequence [53,57], but we do not have experimental evidence of how the CtpA is translocated into the thylakoid membrane.

### 3.5. Catalytic Mechanism of the CtpA

Elucidation of the catalytic mechanism(s) of the CtpA was approached by alanine-scanning mutagenesis [72] and structural analysis [73], which revealed that the CtpA is an unconventional serine protease, which catalytic center consists of serine and lysine dyad (serine provides nucleophile and lysine acts as a general base) [74,75]. The alanine-scanning mutagenesis of *Synechocystis* 6803 CtpA was performed on conserved hydrophilic amino acid residues among the CtpA/Prc/Tsp proteases. In this analysis, Ser313Ala and Lys338Ala (catalytic dyad residues in the *Synechosystis* 6803 CtpA) mutations completely abolished the CtpA activity [72]. In structure of *Scenedesmus* CtpA, unfortunately, Ser372 and Lys397 (catalytic dyad residues in the *Scenedesmus* CtpA) are roughly close, but the distance between the residues is far for linking hydrogen bond directly, which may be in a resting mode [73].

Alanine substitutions of a few other residues (Asp253, Arg255 and Glu316 in the *Synechosystis* 6803 CtpA) also resulted in lost CtpA activity [72]. Among the three, Glu316 is assigned near the catalytic dyad residues in structure of the *Schenedesmus* CtpA (Glu375 in *Schenedesmus*) [73]. The Glu316Gln lost the CtpA activity, while the Glu316Asp remained active, suggesting that carboxyl group on the side chain at this position is important to maintain the CtpA activity [72]. In the analyses of β-lactamase and Prc/Tsp, which are hydrolases belong to the same catalytic mechanism group, mutations at Glu166 of β-lactamase and Asp441 of Tsp, indicated similar behavior in each activity [76,77]. The catalytic involvement of these acidic residues near the catalytic dyads is still under debate and continuous consideration of the catalytic mechanism needed.

### 3.6. Substrate Recognition of the CtpA

For elucidating the substrate specificity of spinach CtpA, a series of in vitro analyses has been conducted using synthetic peptides which have an amino acid substitution on the substrate [55,78,79]. These results revealed that the spinach CtpA has strict substrate specificity, in which the CtpA recognizes pD1 sequence, -DLA:AI- (from P3 to P2′; Asp342-Ile346 of pD1). Among them, the most important residue on the substrate is Leu343 (P2). Spinach CtpA could not cleave the Leu343Ala peptide [78], furthermore, this peptide did not function as a competitive inhibitor [79], which means that the peptide with the Leu343Ala substitution cannot interact with the substrate recognition site of the spinach CtpA. On the other hand, Asp342 (P3) showed a different aspect. The Asp342Asn peptide could not be cleaved, however, this peptide behaved as a competitive inhibitor [55,78]. This result indicates that this peptide interacts with the substrate recognition site in the same manner of the substrates. Surprisingly, the Asp342Glu peptide returned to the substrate again [55], which implies that side chain carboxy group at P3 position of the substrate is crucial for the catalysis, like a substrate-assisted catalysis [80]. This clue that seems to hide any important meanings, therefore, should be studied further. The P1′ and P2′ positions on the substrate are not strictly important for substrate recognition of the spinach CtpA, but substitutions at P1′ position affected Vmax of the cleavage reaction, while substitutions at P2′ increased Km of the substrate recognition [78,79].

To evaluate these substrate specificities in vivo, experiments using *Chlamydomonas reinhardtii* were designed, in which mutations were introduced into the P1′ position (Ser345 of *Chlamydomonas* pD1). Under a pure culture condition, these mutations (Ser345Phe/Val/Gly/Cys) did not affect D1 processing, photosynthetic activity and photoautotrophic growth capacity [81], however in mixed culture conditions, Ser345Cys mutant eradicated Ser345Gly mutant and Ser345Phe mutant overwhelmed Ser345Val mutant, which results indicate that substitutions that give higher Vmax in vitro [78], provides higher competitiveness in photoautotrophic growth in vivo [82].

The CtpA has a notable domain, PDZ, which is protein-protein interaction module that binds to the C-terminus of target proteins [83,84,85]. This character is reasonable for the CtpA that recognizes and cleaves off the C-terminal extension. PDZ domains are found in several proteases not only CtpA homologs, Tsp/Prc [48,49] and CtpB [61], but also HtrA/DegP family members [86,87]. PDZ domains in some proteases are verified to promote protease activity after substrate binding [88,89,90,91,92,93], especially, molecular details of activation mechanism were reported in Tsp/Prc and CtpB [92,93]. Substrate binding to the PDZ domain triggers a structure shift of the Prc/Tsp from the resting state to the activated state. In this shift, the catalytic dyad residues (Ser452 and Lys477 of the Prc/Tsp) approach to hydrogen bond distance each other [92,93]. The *Scenedesmus* CtpA structure [73] is probably in the resting state, substrate binding with its PDZ domain is likely to trigger such structure shift. To define the roles of the PDZ domain of the CtpA in substrate recognition as well as catalytic regulation, further studies are required. Similarly, to clarify accurate substrate recognition manners of the spinach CtpA observed in vitro, structure analysis of the CtpA with substrates (or competitive inhibitors) through co-crystallization must be required, because these results can directly display all contacts between the substrate and the CtpA.

## 4. Efforts for Understanding Significance of the D1 Processing

### 4.1. Issues Raised from Organisms Lacking C-Terminal Extension

This review has frequently mentioned the importance of D1 processing for construction of the WOC, but surprisingly, Keller and Stutz [94] reported that *psbA* gene in *Euglena gracilis* encodes D1 protein lacking C-terminal extension, which means D1 processing is unnecessary for oxygenic photosynthesis in *Euglena*, even though this organism has active PSII [95] and photoautotrophic growth competence [96]. Inspired by this observation, a stop codon was introduced into P1′ site (345th codon) of the D1 protein in *Chlamydomonas* and *Synechocystis* 6803. These mutant strains with no C-terminal extension indicated a normal PSII activity and a comparable photosynthetic growth ability [30,97,98]. It apparently suggests the C-terminal extension of the pD1 is dispensable for formation of a functional PSII. However, this interpretation is difficult to explain that almost all oxygenic photosynthetic eukaryotes and all cyanobacteria have inherited the D1 processing for several billions of years even that the D1 processing forces these organisms to pay extra costs, such as synthesis of the extension and preparing the specific protease CtpA to remove the extension.

Ivleva et al. [99] designed excellent mixed culture analyses to estimate benefit provided from maintaining the D1 processing. When a mixed culture of the normal pD1 strain and extension-less (Ser345stop) strain was conducted under photoautotrophic growth conditions, the normal pD1 strain eradicated the extension-less strain, despite both were able to coexist under heterotrophic growth conditions. When another mixed culture experiment of the normal pD1 strain and a mutant strain having a longer extension, which consists of 29 residues, was performed under photoautotrophic growth conditions, the normal pD1 strain eradicated the longer extension strain. When the extension-less strain and the longer extension strain are grown well under pure culture, both strains indicate comparable photoautotrophic growth competence and photosynthetic activity with those of the normal pD1 strain. These results reveal that the D1 processing provides a very slight but certain benefit for fitness under photoautotrophic growth conditions. Kuvikova et al. [100] reported an increase of unassembled D1 protein into PSII complex in the extension-less mutant, which suggests that the C-terminal extension mediates the efficient assembly of the newly synthesized D1 protein into PSII. In addition, Kuvikova et al. also proposed that Asn359 of pD1 has some buffer actions against photoinhibition [100]. Existence of the C-terminal extension is likely to supply slight merit in the repair process from photoinhibition, especially in the assembly step(s) of the newly synthesized D1 protein into PSII (see also Section 4.3 and Section 4.4).

Until now, several organisms lacking C-terminal extension like *Euglena* have been reported, however they were only detected in limited species of secondary or tertiary symbiotic algae, Euglenophyceae, Chlorarachniophyceae and Dinophyceae [20,31,32,33]. Since these organisms appear on various branches of the phylogenic tree [32,33,101], this trait has been independently acquired by convergent evolution. During endosymbiotic gene transfer from the symbiont to nucleus of the host [31,102], these organisms might have opted for loss of the C-terminal extension, instead of transfer of the *CtpA* gene.

### 4.2. Significance of C-Terminus for the WOC Construction

The CtpA deficient mutant LF-1 has widely contributed to make a consensus that D1 processing is crucial for the construction of the WOC. However, in 1980s–90s, molecular biological manipulation of *Scenedesmus* was difficult, so the center of research shifted toward a model organism, *Synechocystis* 6803.

The introduction of a stop codon at P1′ site (Ser345stop) of D1 protein in *Synechocystis* 6803 is allowed to grow photoautotrophically, however, stop codons introduced at further upstream of the D1 protein, such as Asn335stop, Asp342stop, Leu343stop and Ala344stop, disrupted photoautotrophic growth competence [30]. Although all the mutations described in this section lost the C-terminal extensions, only the mutation, Ser345stop, is allowed the photoautotrophic growth. This result presumes that the C-terminal extension do not inhibit construction of the WOC by steric hindrance, but rather the free carboxyl group of main chain of the Ala344 exposed after cleavage is important to ligate Mn atom(s) of the Mn_4_CaO_5_ cluster [30]. Based on the concept, mutants in which Ala344 of D1 protein was replaced with various amino acid residues were constructed from a *Synechocystis* 6803 Ser345stop mutant. Using the Ser345stop mutant as the host strain allowed the experiments to disconnect from the influence of the CtpA cleavage. Evaluation of these mutants by isotope-edited Fourier transform infrared (FTIR) spectroscopy, EPR, thermoluminescence, ^13^C- electron nuclear double resonance (ENDOR) and metal substitution analyses indicated that the C-terminus of D1 protein is a ligand for a Mn atom in the Mn_4_CaO_5_ cluster [103,104,105,106,107,108]. Furthermore, Consideration from changes of these signals in the S-state cycle (Kok’s cycle) [1] suggested that the C-terminus of D1 protein ligates a Mn atom which is oxidized from Mn^3+^ to Mn^4+^ during the S_1_ to S_2_ transition and is reduced to Mn^3+^ during the S_3_ to S_0_ transition through S_4_ [103,106].

With the remarkable progress of structure analysis in recent years, we can refer the structure of the PSII isolated from thermophilic cyanobacteria (*Thermosynechococcus elongatus*) with a resolution of 1.9 Å [109]. In addition, the radiation damage free PSII structure has been solved by femtosecond X-ray free electron lasers (XFEL) [110]. In the structure, the oxygen atoms in carboxyl group of Ala344 (C-terminal residues of D1 protein) are assigned to a distance of 1.9 Å with Mn2C atom and a distance of 2.43 Å with Ca atom in the Mn_4_CaO_5_-cluster. An FTIR analysis performed using the Ala344 mutants with metal substitution from Ca to strontium showed that the C-terminal carboxyl group of D1 protein do not coordinate Ca atom [107], but this interpretation is inconsistent with the structure [6,109,110]. Finally, structure data clarify that in organisms holding the C-terminal extension, D1 processing is absolutely required to expose the main chain carboxyl-group of Ala344 for construction of the WOC.

Construction of the WOC of the PSII proceeds with the two-quantum mechanism which consists of two photon reactions with a connecting dark rearrangement period. The photon reactions are tightly linked with charge separation of P680, therefore the process is called photoactivation [5,111]. Thanks to the enormous efforts of many researchers, the molecular details of the photoactivation are beginning to be elucidated [111,112,113,114]. However, the photoactivation has been studied in vitro as a regenerating process of the WOC of the PSII in which the Mn_4_CaO_5_ cluster is pre-removed by chemical treatment. These results are certain and might contribute to elucidate the photoactivation in vivo, but at this stage, it is not guaranteed that photoactivation processes performed in vivo and in vitro are the same. An attempt to simulate the in vivo photoactivation process of the WOC, which was imported by the structural data, the kinetics of Mn-Ca assembly and the effects of D1 processing was conducted [115]. Since this research, various new results have been accumulated, further attempts with these novel data are awaited.

### 4.3. D1 Processing and Assembly Supporting Factors of PSII

When considering the PSII assembly, we must distinguish de novo assembly and repair assembly. However, the boundary between them is still unclear [15]. This review primarily concentrates on D1 processing during the repair process of PSII, but and inevitably contains some explanations of D1 processing in de novo assembly.

D1 processing is repressed under various stress conditions. Low temperature slows down the D1 processing in *Synechocystis* 6803, furthermore this phenotype is promoted in the desaturase deficient mutant in which fluidity of thylakoid membranes is reduced [116,117]. Light stress, such as prolonged strong light exposure to *Synechocystis* 6803 also leads the similar phenotypes, repression of D1 processing and its promotion in the desaturase mutant [118,119]. In addition, D1 processing in *Synechocystis* 6803 is also repressed by interruption of chlorophyll supply [120]. These early observations imply the presence of some factors other than the CtpA which affects the pD1 processing.

Early research of PSII assembly begun by pursuing steady subunits of PSII in pulse-chase experiments on the two-dimensional gel electrophoresis that combines native-PAGE and SDS-PAGE [121]. Newly synthesized D1 protein is translated by thylakoid bound ribosomes and incorporated into thylakoid co-translationally under supports of SRP-SecY pathway [122], which takes place in CP43/D1-free PSII (Figure 3[I]). The CP43/D1-free PSII is an assembly intermediate in the PSII repair cycle which appears after separation of CP43 and proteolytic removal of photodamaged D1 protein [16]. After insertion of the newly synthesized pD1, CP43 is reassembled, pD1 is processed (Figure 3[II]) and finally photoactivation of the WOC is accomplished (Figure 3[III]). These PSII intermediates do not bind the PsbO/PsbP/PsbQ peripheral proteins (PsbO/PsbU/PsbV/PsbQ for red lineage), therefore, CtpA can approach the C-terminus of pD1 to cleave off the extension (Figure 3[II]) [16].

In the last 20 years, the various assembly supporting factors which are not steady subunits of the PSII, but significantly influence PSII assembly, have been discovered and characterized [123,124]. Notably, these assembly supporting factors have been identified not only from *Synechocystis* 6803, but also from higher plants such as *Arabidopsis thaliana*, through characterization of photosynthetic deficient mutants, *low PSII accumulation* (*lpa*), *high chlorophyll fluorescence* (*hcf* ) and *photosynthesis affected mutant* (*pam*) [123,124].

Processing-associated tetratricopeptide repeat protein: slr2048 (PratA) of *Synechocystis* 6803 which is a such regulatory factor, tightly relates with D1 processing [125]. In the *pratA* mutant, PSII content was drastically reduced and D1 processing was repressed. The PratA that is a lumenal peripheral protein, has ability to bind C-terminal sequence of the D1 protein [125,126], which implies that the factor holds some roles over the D1 processing and/or construction of the WOC. Stengel et al. proposed that the PratA is a Mn^2+^ binding protein and probably coordinates efficient delivery of Mn atoms to PSII for supporting construction of Mn_4_CaO_5_ cluster, however localization of the PratA is highly restricted at cell periphery, which may mean that the PratA mediates de novo assembly of PSII [127]. Characterization of an *Arabidopsis* homolog, LPA1 revealed that the factor certainly mediates efficient assembly of PSII, however no repression of the D1 processing was detected in the *lpa1* mutant [128].

Psb27 (slr1645) of *Synechocystis* 6803 was identified as a putative subunit of PSII by proteome analysis of PSII complex [129], but it is not a steady subunit of PSII [6]. Interestingly, the Psb27 was abundantly detected in PSII complex isolated from the *ctpA* deficient mutant, instead of peripheral proteins PsbO/PsbU/PsbV/PsbQ (Figure 3[II]) [130]. LPA19, a Psb27 homolog in *Arabidopsis thaliana*, correlates with D1 processing, because D1 processing was highly repressed in *lpa19* mutant [131]. As mentioned, the Psb27 has a characteristic suitable as a key factor for D1 processing during assembly of PSII, so I would like to mention it in detail in the next section.

HCF136 is an assembly supporting factor for PSII, identified by characterization of an *Arabidopsis* mutant, *hcf136* which was devoid of any PSII activity [132,133]. YCF48 (slr2034), a *Synechocystis* 6803 homolog of HCF136 interacts with C-terminal extension of pD1 [134], which interaction may stabilize pD1 protein on thylakoids until assembling with its counterpart, D2 protein during de novo assembly of PSII [135]. From the structural insight of the HCF136/YCF48, a seven-bladed β-propeller structure, Yu et al. [136] proposed a model in which the Ycf48 protein holds C-terminal extension of pD1 to escort incorporation of chlorophylls into the correct positions of the newly synthesized D1 protein. This model may relate with an observation that restriction of chlorophyll supply retarded the D1 processing [120].

Light irradiation forms a proton gradient across thylakoid membranes coupled with photosynthetic electron transports, which results that lumen matrix turns to acidic pH [1,137]. Synthesis of the D1 protein is activated in the light, and then D1 processing is frequently performed under such conditions. However, pH optimum of the spinach CtpA activity obtained by in vitro experiments is 8.0 [138], unsuitable for the enzyme acting in the lumen matrix in the light. Since this value was obtained in vitro experiments using synthetic peptide or cell-free translated pD1 as substrates, pH optimum of the spinach CtpA activity was re-evaluated using pD1 embedded in thylakoids isolated from the LF-1 mutant as a substrate. Surprisingly, using a substrate closer to in vivo, the pH optimum of the spinach CtpA is shifted to pH 6.0, a reasonable value for enzymes acting in lumen matrix [138]. The CtpA and/or pD1 interact(s) with several assembly supporting factors, which probably exist on the LF-1 thylakoid membranes and these interactions may shift the pH optimum. Further studies are required on what interactions cause the pH optimum shift.

### 4.4. Structural Appoach for Understanding the D1 Processing during PSII Assembly

As mentioned above, PSII complex isolated from the *ctpA*-deficient mutant of *Synechocystis* 6803 contains the Psb27 protein, instead of the peripheral proteins PsbO/PsbU/PsbV/PsbQ (Figure 3[II]). Since this complex contains Psb27, an assembly supporting factor, and pD1, it is considered to be an assembly intermediate. However, PratA and YCF48 could not be detected in the intermediate [130]. Pakrasi’s group sought to examine what intermediates would appear in the *psb27* deficient mutant, but surprisingly the mutant of *Synechocystis* 6803 indicated normal PSII activity and photoautotrophic growth competence. Even with the loss of the Psb27, PSII can be assembled normally, which is apparently similar with the averting mechanism in deletion of the C-terminal extension described in Section 4.1. After repeating various conditions, the mutant indicated very slightly weakened recovery from the photoinhibition and very slightly lowered photoactivity efficiency of the WOC [139]. In the mixed culture experiment of wild type strain and the *psb27* deficient strain, a severe condition, such as using a CaCl_2_-free medium, is required to eradicate the *psb27* deficient strain [139]. CaCl_2_, both elements are essential for the photoactivation and maintenance of the WOC, and removal of the both made a very slightly difference in photoautotrophic growth between the mutant and wild type strain [139]. The Psb27 is a soluble protein with lipid modification [140] which character may support binding to lumen side of CP43 (loop E) in the PSII assembly intermediate (Figure 3[II]) [141,142,143]. These accumulated results suggest that the Psb27 prevents the binding of peripheral proteins PsbU/PsbV/PsbQ to the PSII assembly intermediate, which secures the route for approach of the CtpA to C-terminus of pD1 in the assembly intermediate (Figure 3[II]). After removal of the C-terminal extension, loop E of CP43 in the intermediate is exposed to the lumen matrix, which is caused by release of the Psb27 from the intermediate [143], and resultant structure changes may allow to bind PsbO to the assembly intermediate [142]. These observations suggest that the peripheral proteins PsbO/PsbU/PsbV/PsbQ bind stepwise according to the states of the assembly intermediates (Figure 3[III]).

In recent years, significant development of cryo-electron microscopy techniques opens the structural analysis of such assembly intermediates. According to the concept, several groups reported structures of putative assembly intermediates of PSII [144,145]. However, this technique has a weak point to solve redox-active metal-proteins like PSII, because electron beams inevitably reduce the metal atoms, which sometime provides non-negligible errors on positions of metal atoms [146].

Using cryo-microscopy, Zabret et al. solved a *T. elongatus* PSII structure at 2.94 Å resolution isolated from *psbJ*-deficient mutant, which PSII complex contains three assembly supporting factors, Psb27, Psb28 and Psb34 [145]. In this structure, Psb27 is assigned on the lumen side of CP43 as predicted biochemical analyses [141,143]. However, the structure indicates that Psb27 can not physically interfere with the binding of the peripheral proteins PsbO/PsbV/PsbU, because the binding site of Psb27 does not have significant overlap with the binding sites of these proteins. We have to find another reason why the peripheral proteins cannot bind the complex. In addition, the Psb27 in this structure has no direct contact to the pocket for the Mn_4_CaO_5_ cluster as well as the C terminus of the D1 protein. Furthermore, in this structure, D1 protein is already processed, C-terminus of D1 protein is assigned at a different position compared with the reported PSII structure [6,109,110], Mn_4_CaO_5_ cluster is not constructed and peripheral proteins PsbO/PsbV/PsbU do not bind, which findings are slightly distant from our perception of assembly intermediates [16]. Further investigations are required whether this structure corresponds a natural assembly intermediate or not. In parallel, Xiao et al. also reported a *T. elongatus* PSII structure isolated from His-tagged Psb28 and *psbV* deficient strain using cryo-microscopy [144]. In this structure, C-terminus of D1 protein could not be assigned, no Mn_4_CaO_5_ cluster and this complex does not have the Psb27, which observations are also far from our knowledge of assembly intermediates. We must dig into the relationships between these structures and natural assembly intermediates. Therefore, unfortunately, these structures cannot provide any information about relationships between D1 processing and the WOC construction during the PSII repair process, but the potential of this approach is worth enough. Therefore, further analyses to solve structures of PSII assembly intermediates with pD1 or a natural intermediate during construction of the WOC are awaited.

## 5. Remaining Mysteries in the D1 Processing

### 5.1. Evolution of the C-Terminal Extension: Which Is the Prototype? How Did It Evolve?

As mentioned, lengths and sequences of the C-terminal extension have some variety. In this section, I concentrate on the variations on the lengths of the extension and consider their evolutional relationships. Extension-less organisms (Group III in [20]) found in some algae evolved by secondary or tertiary symbiotic event are excluded from the discussion as exceptions.

Chlorophyll *b* containing organisms (Group I in [20]), Chlorophyta, Streptophyta and *Prochlorothrix*, possess the short extension which consists of 8–9 residues (green rounded rectangles in Figure 4). This extension is removed by one step of proteolytic cleavage by the CtpA in all organisms examined to date. On the other hand, organisms on red lineage [147] (Group II in [20]), Glaucophyta, Rhodophyta and most cyanobacteria, have the long extension which consists of 15-16 residues (rounded rectangles with numbers on orange background circles in Figure 4). In *Synechocystis* 6803, the long extension is cleaved off by successive two steps by the CtpA [148]. The second (= final) cleavage is carried out at the canonical site for exposing carboxyl group of Ala344, while the first cleavage site is the peptide bond between Ala352 and Leu353 of pD1 [149]. We do not know whether the successive two step cleavage for removing the long extension is a universal phenomenon in Group II organisms or a phenomenon specific for *Synechocystis* 6803. Assuming that the successive two step cleavage is a universal phenomenon in Group II organisms and evolution is directing toward increasing complexity, the short extension is simple and is more likely to be a prototype. Whichever is the prototype, it is a complete black box as to how it diverged and how it evolved into the current form.

In the past, the variety of manners on D1 processing caused a confusion. The groups I and II, both include prokaryotes and eukaryotes, furthermore strong continuity is detected between prokaryotes and eukaryotes within each group. Based on the observation, it was once proposed that the primary symbiosis for generating chloroplasts was carried out on each group independently [150]. This hypothesis has been denied through the refinement of the evolutionary tree, which forms the current consensus that only one primary symbiotic event rose to all the chloroplasts [151]. Probably, this confusing complexity across the groups, is results raised by horizontal and retrograde gene transfers through competence of cyanobacteria to acquire DNA from the environment [44] or infection of cyanophage [152,153,154]. Thus, it is completely unknown the prototype of the D1 processing and how it evolved to the current styles.

To obtain a clue approaching the prototype of the D1 processing, the chance may be brought from the identification of cyanobacteria involved in the primary symbiosis. For example, based on the report by Ponce-Toledo et al. [155], *Gloeomargarita lithophora* is the closest extant relative of the plastid ancestor, which D1 protein has 12 residues long C-terminal extension (GenBank Acc. No. CP017675). On the other hand, Moore et al. [156] reported that the extant closest with the plastid ancestor is *Synechococcus* sp. JA-2-3B’a(2-13) which genome has at least two *psbA* genes, and one of them encodes the 10 residues long C-terminal extension (NCBI Reference Sequence: WP_011432036.1). Both cyanobacteria have exceptionally short C-terminal extensions (Figure 4), which seems meaningful, however, careful consideration is required understanding these observations. (Nota bene 3: Currently, a rhizarian amoeba *Paulinella* has an independently obtained primary plastid, which symbiotic event began approximately 120 million years ago [157]. This event resulted to obtain a plastid like photosynthetic organelle and still continues, for example, the *ctpA* genes is sitting on the plastid genome (GenBank Acc. No. KX897545.1, KY124271.1 and MG976688.1). Therefore, this review focuses only on the “canonical” chloroplasts).

### 5.2. Approaching to Merits Given by D1 Processing: What Are the Benefits?

Mixed culture experiment revealed that D1 processing supplies slight but certain merits to oxygenic photosynthetic organisms for surviving wisely [99], however, we do not know the actual mechanisms of the merits. Based on a concept that the C-terminal extension carries any signals for supporting PSII assembly, Kuvikova et al. [100] constructed and evaluated site-directed mutants in which few residues on the C-terminal extension were substituted. Their results suggested that Asn359 carries information that provides some benefits in repair process from photoinhibition. However, other parts have not been investigated at all. To approach the signals on the C-terminal extension, I think that application of protein-engineering technology, such as iterative saturation mutagenesis and adaptive laboratory evolution, is a good option [158,159,160,161]. After laboratory evolution under appropriate design and conditions, if sequence convergence can be detected on the C-terminal extension, which proves that the C-terminal extension carries some signals supplying benefit for surviving these conditions. Even if such a result is obtained, it is required to pay attention to secondary effects, such as mutations on the genome induced for compromise to coexist with the converged extension sequence.

At the end of this review, I propose a novel hypothesis that may explain significance of the D1 processing. This hypothesis is based on a fact the most important function of D1 processing is exposing of the carboxyl group of Ala344. This process can evade through replacing the codon at position 345 with stop codon, but most oxygenic photosynthetic organisms have not chosen this way. In this context, my hypothesis further assumes a fact that translation termination efficiency is not 100% [162], which should act as a selection pressure for the protein that requires accuracy of translation termination, like D1 protein. Some readthroughs inevitably occur at native stop codons, especially under stress conditions [163,164]. D1 protein is massively synthesized and processed under light condition, in which translational equipment is exposed to ROS, heat and high stromal pH. In fact, singlet oxygen, one of the ROS, worsens elongation steps of the D1 protein translation in *Synechocystis* 6803 [165]. My hypothesis “precise cut after designed readthrough” seems to be a desirable option to accurately expose the C-terminus of D1 protein in the presence of accidental readthroughs. Moreover, this “precise cut after designed readthrough” phenomenon has compatibility with placing some signals for supporting PSII assembly onto the C-terminal extension, which provides more advantage to the organisms and then promotes the fixation of this process.

The ultimate origin of the D1 processing may be related to the following phenomenon. Organisms including cyanobacteria, furnish the rescue system for ribosomes stalled by an accidental translation interruption, in which a small RNA termed tmRNA acts as a key player [166,167,168]. In this system, the tmRNA enters the A site of the stalled ribosome like a tRNA, and mRNA like region of the tmRNA is read instead of the stalled mRNA, which resultantly adds a common tag-sequence to the C-terminus of the aberrant peptide (this system also called *trans*-translation) [166,167,168]. The tag-sequence in *E. coli*, which sequence is 10 residues long, is AANDENYALAA. Keiler et al. [169] proposed that the Prc/Tsp protease is involved in degradation of some tagged proteins generated by the tmRNA mediated ribosome rescue system. This system, although it does not act in the translation termination, may be linked with the origin of the D1 processing at the deep root. Further experimental verification is required and awaited to prove this hypothesis.

## 6. Perspectives

As mentioned, research on D1 processing has been conducted by many techniques based on genetics, biochemistry, molecular biology, physical chemistry and structural biology. Especially, the resolution of the PSII structure has been significantly improved and now we have 1.9 Å resolution structure data of PSII. These structures clearly indicate that C-terminus of D1 protein ligates Mn and Ca atoms in the Mn_4_CaO_5_ cluster of the WOC of PSII. These results helped us to appreciate the significance of the D1 processing as a crucial step to expose the C-terminus of the D1 protein in organisms that have C-terminal extension. However, big mysteries are still remaining on the emergence, evolution and merits of the D1 processing. Since almost all extant cyanobacteria inherit the D1 processing, the phenomenon has emerged in a primitive cyanobacterium, but elucidation of the prototype and clarifying the paths from the prototype to the current variations that extended to eukaryotes, remain a challenge. Moreover, the D1 processing definitely provides oxygenic photosynthesis organisms with some benefits worth keeping for several billions of years, however the molecular mechanism of this merit is completely unknown. Solutions of these issues are strongly awaited.

On the other hand, the significant development of cryo-electron microscopy techniques opens the possibility of structural analysis of assembly intermediates of the PSII. With such excellent technology, we are on the verge a great opportunity to clarify the relationships among PSII assembly, D1 processing, and photoactivation events at molecular level. Therefore, actions of the D1 processing at molecular level during the PSII assembly is still unclear, but by application of the new technologies, we may reach the truth sooner than expected.

Currently, the number of researchers specializing in the D1 processing is decreasing. I think to need that more researchers from different fields who will supply new perspectives, will participate in elucidation of the D1 processing. I want that this review will help these newcomers or re-visitors to touch and organize the current core of the D1 processing research.

## Figures and Tables

**Figure 1 ijms-23-02520-f001:**
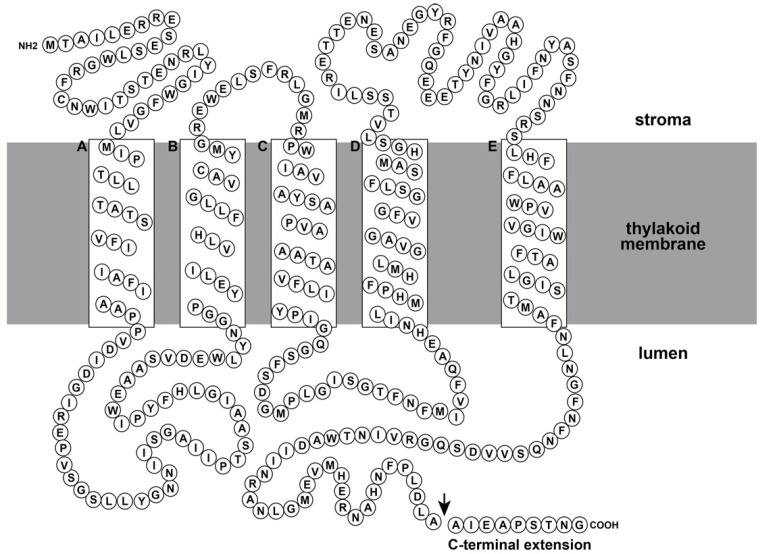
Predicted topology of the spinach D1 protein in 1980s. This topology model is basically rational, even now that the molecular structure has been revealed [6]. The figure is drawn based on the model reported by Trebst [64] with minor modifications. The sequence of D1 protein is indicated by one letter abbreviation of amino acids. White boxes represent predicted alpha-helices for transmembrane. An arrow indicates the cleavage site by the CtpA.

**Figure 2 ijms-23-02520-f002:**
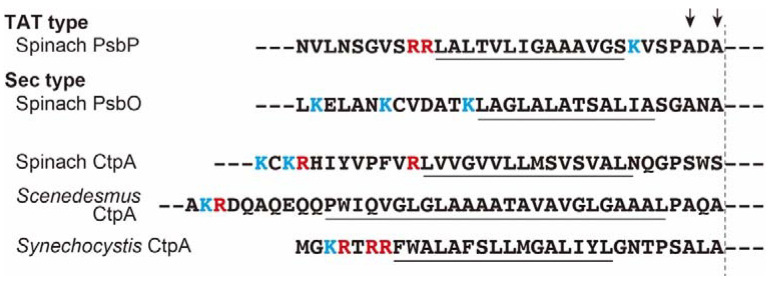
Amino acid sequences of signal peptides of the CtpAs (spinach, *Scenedesmus* and *Synechocystis* 6803) for targeting to the thylakoid. Sequences of the typical signal peptides for Tat-pathway (spinach PsbP) and Sec-pathway (spinach PsbO) are also aligned. Arginine and lysine residues which characteristically appear in these signals, are shown in red and blue, respectively. Hydrophobic domains are underlined. Downward arrows indicate residues under the “-3, -1 rule” of von Heijne [70,71] for the cleavage of the signal peptidase (thylakoidal processing protease) and the vertical broken line means cleavage site by the signal peptidase.

**Figure 3 ijms-23-02520-f003:**
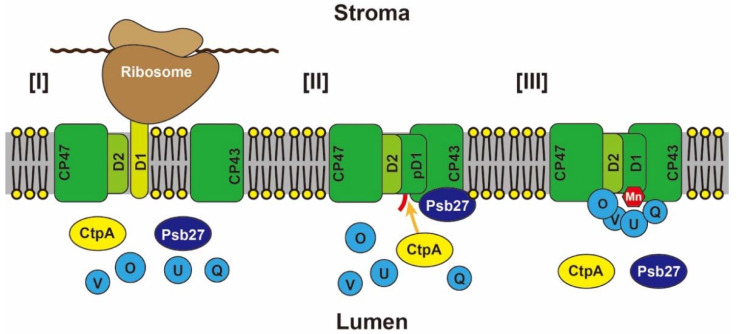
Schematic model of a part in the PSII repair cycles from the D1 protein synthesis [I] to the WOC formation [III] through the D1 processing by the CtpA [II]. This model is written primarily from results obtained with *Synechocystis* 6803, and contains only the key supporting factors/subunits of the PSII during the assembly discussed in the text. Red tail hanging on the pD1 in [II] indicates the C-terminal extension. A red hexagon indicates Mn cluster of the WOC.

**Figure 4 ijms-23-02520-f004:**
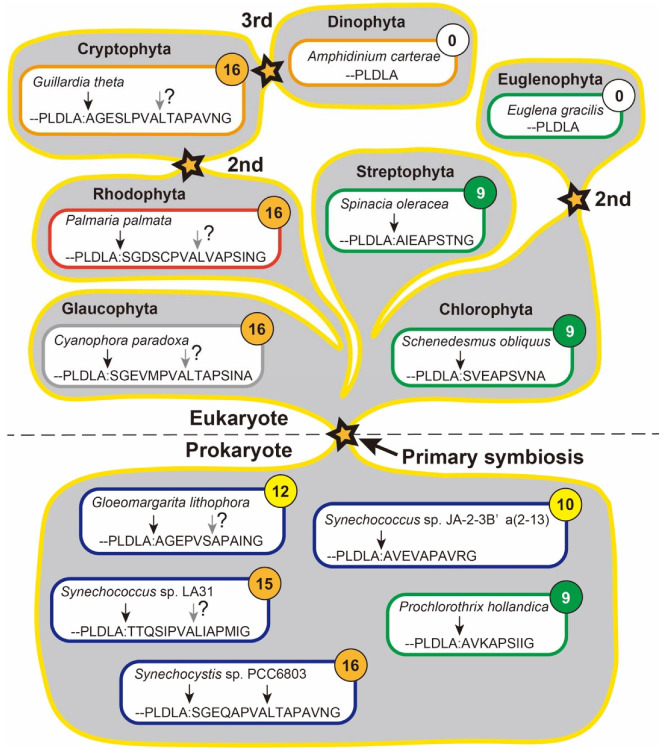
Variations on the C-terminal extension sequences and distribution of the variety on the schematic tree of the evolutionary history of plastid-bearing organisms. Rounded rectangles represent typical organisms in the groups. The scientific names of each organism are shown in the upper row, and its C-terminal sequence of pD1 is indicated in the lower row. The downward arrows mean the cleavage site of the CtpA. Gray arrows with question marks indicate potential cleavage sites without experimental confirmation. The numbers in the circles of each organism are lengths of the C-terminal extension. Stars mean symbiotic events.

## Data Availability

Not applicable.

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
