# Peer review of "Processing of D1 Protein: A Mysterious Process Carried Out in Thylakoid Lumen"

_ijms, 2022, doi:10.3390/ijms23052520_

Round 1

Reviewer 1 Report

This review summarizes the research associated with the processing of the C-terminal extension of the D1 protein in the thylakoid. The review offers a valuable and comprehensive overview of the role of this posttranslational modification in the assembly of the water-oxidizing complex (WOC). This analysis is performed from different perspectives, including a structural, functional, and evolutionary point of view.  The manuscript can be published in its current form.

Author Response

Response to Reviewer 1 Comments

This review summarizes the research associated with the processing of the C-terminal extension of the D1 protein in the thylakoid. The review offers a valuable and comprehensive overview of the role of this posttranslational modification in the assembly of the water-oxidizing complex (WOC). This analysis is performed from different perspectives, including a structural, functional, and evolutionary point of view.  The manuscript can be published in its current form.

Ans: Thank you very much for your careful reviewing.

I revised the chapter 3.4. Localization of the CtpA to respond to the comments    the reviewer 3.

I would appreciate if you could confirm this revision.

Reviewer 2 Report

 In the manuscript(ijms-1575330) entitled "Processing of D1 protein; a mysterious process carried out in 2 thylakoid lumen", the author reviewed the proteolytic processing of D1 protein, a core subunit of photosystem II by a specific serine protease, CtpA. The manuscript reviewed mainly CtpA along with the 4 points, discovery of the processing and its importance, the protease responsible for the processing, significance and unrevealed points in D1 processing. The manuscript  reviewed well and precisely  about the investigation of the processing. The review contributes to further research of the unrevealed point of the processing mechanism and the construction of photosystem II.

  So, I recommend that the manuscript be accepted for publication in International Journal of Molecular Sciences. 

Author Response

 In the manuscript(ijms-1575330) entitled "Processing of D1 protein; a mysterious process carried out in thylakoid lumen", the author reviewed the proteolytic processing of D1 protein, a core subunit of photosystem II by a specific serine protease, CtpA. The manuscript reviewed mainly CtpA along with the 4 points, discovery of the processing and its importance, the protease responsible for the processing, significance and unrevealed points in D1 processing. The manuscript  reviewed well and precisely  about the investigation of the processing. The review contributes to further research of the unrevealed point of the processing mechanism and the construction of photosystem II.

  So, I recommend that the manuscript be accepted for publication in International Journal of Molecular Sciences. 

Ans: Thank you very much for your careful reviewing.

I revised the chapter 3.4. Localization of the CtpA to respond to the comments    the reviewer 3.

I would appreciate if you could confirm this revision.

Reviewer 3 Report

This is a nice review on the photosynthetic D1 protein and its processing protease CtpA. I have enjoyed reading it but I think there are some issues that could improve this review. A major focus is the processing of the protein during its biosynthesis. However, too little information is given to the N-terminal processing steps. Already in the introduction (line 64 ff) it should be clearly stated that there is also an N-terminal processing to target the protein to the chloroplast and thylakoid.

More detaills should be provided in localization chapter (line 18 ff). Is the 150 residue long signal peptide a bipartite presequence? Which proteases are involved here and where are they localized?

In line 156 it should be clearly mentioned that D1 is a membrane protein that spans the thylakoid membrane multiple (?) times. A schematic of the protein topology would be helpful. 

The chapter on the targeting mechanism (line 193 ff) is not really convincing. The only clear TAT signal is the spinach PsbP protein and maybe the Synechocystis CtpA. It is known that one R residue is not sufficient for Tat. The reasons for proteins to use TAT should be mentioned. This is when they bind cofactors as folded proteins in the cytoplasm. Is this the case for the proteins listed? One drawback of TAT is that its secretion rate is very low compared to Sec.

Some small errors:

line 152:  "promoter"

line 237: omit "should be"

line 334:  "extensions"

line 337: what are the "various" Ala mutants?

line 352:  "a distance of"

line 369: imported by the structural

 line 373: omit "into"

line 427: "suitable as a key factor"

line 444: "obtained by in vitro experiments"

line 448: , a reasonable value... 

line 510:  "Remaining"

Author Response

Response to Reviewer  3 Comments

This is a nice review on the photosynthetic D1 protein and its processing protease CtpA. I have enjoyed reading it but I think there are some issues that could improve this review.

Ans: Thank you very much for your careful reviewing. I could recognize the missing pieces of my manuscript.

A major focus is the processing of the protein during its biosynthesis. However, too little information is given to the N-terminal processing steps. Already in the introduction (line 64 ff) it should be clearly stated that there is also an N-terminal processing to target the protein to the chloroplast and thylakoid.

Ans: I understand the meaning of your suggestion. However, I think that it is difficult to reveal the clear conclusions in the Introduction section, because such information seems to interfere with the reader’s  tracing the history of research in order. Instead, I've revised and elaborated on the chapter 3.4 (localization).

More details should be provided in localization chapter (line 18 ff). Is the 150 residue long signal peptide a bipartite presequence? Which proteases are involved here and where are they localized?

Ans: I tried to elaborate more detail to explain N-terminal pre-sequence of the CtpAs. However, there are many issues that have not been clarified experimentally.

In line 156 it should be clearly mentioned that D1 is a membrane protein that spans the thylakoid membrane multiple (?) times. A schematic of the protein topology would be helpful. 

Ans: I insert the new Figure 1 to show the protein topology of D1 protein.

The chapter on the targeting mechanism (line 193 ff) is not really convincing. The only clear TAT signal is the spinach PsbP protein and maybe the Synechocystis CtpA. It is known that one R residue is not sufficient for Tat. The reasons for proteins to use TAT should be mentioned. This is when they bind cofactors as folded proteins in the cytoplasm. Is this the case for the proteins listed? One drawback of TAT is that its secretion rate is very low compared to Sec.

Ans: My literature learning was insufficient at the first submission and your exact suggestion has erased my confusion. The result (Karnauchov et al. 1997) that Synechocystis CtpA was partially transported by the TAT pathway in the chloroplasts of higher plants, is understandable, because the pre-sequence of Synechocystis CtpA holds the twin arginine. On the other hand, the pre-sequences of CtpAs of spinach and Scenedesmus do not have two consecutive arginine residues, implying that these CtpAs are not likely to transport through the TAT pathway, as you suggested. The eukaryotic CtpAs probably transport into thylakoid through Sec pathway based on the characteristic of the pre-sequences. However, at this stage, experimental proof should be still required. I revised the manuscript in this direction.

The CtpA does not require any cofactors for the catalytic activity, therefore Synechocystis CtpA dose not have any necessarily to transport into lumen through TAT pathway. In my review, the study by Karnauchov et al is cited as one of the few reports approaching to the transport mechanism of CtpA to thylakoid lumen, but it is a chimera experiment combined cyanobacteria signal and chloroplasts of higher plants, which is essentially difficult to interpret.

In previous studies of CtpA, the localization site of the CtpA has been firmed to be in thylakoid lumen, but the transport route has not been clarified experimentally. For this reason, I am sorry that I cannot answer your questions completely.

Some small errors:

Ans: I revised follows according to your suggestions.

line 152:  "promoter"

line 237: omit "should be"

line 334:  "extensions"

line 337: what are the "various" Ala mutants?

Ans: “mutants in which Ala344 of D1 protein was replaced with various amino acid residues” I revised it like this.

line 352:  "a distance of"

line 369: imported by the structural

 line 373: omit "into"

line 427: "suitable as a key factor"

line 444: "obtained by in vitro experiments"

line 448: , a reasonable value... 

line 510:  "Remaining"